# Dual RNA-seq reveals transcriptome changes during *Fusarium virguliforme-Trichoderma afroharzianum* interactions

Mirian F. Pimentel[1,2], Leonardo F. Rocha[2]*, Arjun Subedi[2], Jason P. Bond[2], Ahmad M. Fakhoury[2]

1 School of Agriculture Sciences, Southern Illinois University Carbondale, Illinois, United States of America,
2 BASF- Global Agricultural Solutions, Durham, North Carolina, United States of America

* leonardo.rocha@siu.edu

**Data Availability Statement:** Sequence datasets were submitted to the NCBI (National Center for Biotechnology Information) Sequence Read

## Abstract

*Trichoderma* spp. are among the most studied biocontrol agents. While extensive work has been done to understand *Trichoderma* antagonistic mechanisms, additional research is needed to fully understand how *Trichoderma* spp. recognize the pathogen-host and the intra-species variability i frequently observed upon interaction with a specific pathogen-host. This study focuses on elucidating the mechanisms underlying observed phenotypic differences among the *T. afroharzianum* isolates Th19A and Th4 during confrontation with *Fusarium virguliforme* by investigating differences in their transcriptome at different stages of interaction. In a dual plate assay, Th19A overgrows *F. virguliforme*, whereas Th4 forms an inhibition zone. Significant differences were observed in the *F. virguliforme* transcriptome upon interaction with Th19A compared to Th4 and across the different stages of interaction. GO molecular function categories enriched for *F. virguliforme* genes differed, indicating possible transcriptional plasticity upon interaction with Th19A versus Th4. Significant transcriptome changes were also observed in *T. afroharzianum*, with several differences in GO-enriched categories between isolates. Several differentially expressed genes-encoding secreted proteins, including CAZymes and CBM1-domain-containing proteins, were up-regulated in Th19A and Th4 upon interaction with the pathogen, even before physical contact, demonstrating possible volatile-mediated recognition of both isolates by *F. virguliforme*. This study contributes to a better understanding of the interaction between *T. afroharzianum* and *F. virguliforme*, which is crucial for developing efficient biological control programs.

## Introduction

Biological control can be a powerful tool to manage plant diseases, especially for pathogens that are difficult to control, e.g. soil-borne pathogens, where current management practices are not fully effective. Still, the broader use of biological control in agriculture, especially in row crops, is hindered by major difficulties. These include the lack of understanding of how complex biotic and abiotic factors interact with biocontrol agents's (BCAs) and the inconsistent

Archive (SRA) and are available under the accession number PRJNA1052097.

**Funding:** The author(s) received no specific funding for this work.

**Competing interests:** The authors declare no conflict of interest.

efficacy of BCAs under different environments and production systems. Another factor would be the limited knowledge regarding the reasons underlying the failure of biocontrol, which can be partially associated with the lack of understanding of the biological and molecular mechanisms governing the BCA-pathogen interaction [1–3]. Assuredly, the success of biological control programs in crop protection relies on a thorough understanding of the biological and molecular mechanisms influencing the interaction between antagonists and target-phytopathogens. Often times the antagonistic activity of BCAs, such as *Trichoderma spp.*, and their resulting disease suppression is caused by the synergistic effect of several mechanisms of action, including mycoparasitism, competition for space and nutrients, antibiosis, induction of plant defenses, and plant growth promotion [4–6]. In the case of *Trichoderma*, even though a fair amount of work has been done to unravel the cellular and molecular basis of the interactions between *Trichoderma* and their targeted hosts [7, 8], there is still much to be learned regarding unique responses from interactions of specific *Trichoderma* strains against particular phytopathogens [9–12]. In a previous study, the antagonistic activity of several unique *Trichoderma* isolates was screened against *Fusarium virguliforme* [13]. Interestingly, phenotypic differences among these *Trichoderma* isolates during confrontation with *F. virguliforme* were observed during *in vitro* dual plate assays. Th19A (*T. afroharzianum* KMISO2-2-19A) overgrew *F. virguliforme* completely whereas Th4 (*T. afroharzianum* WMNSO2-4-4) formed an inhibition zone. However, both isolates inhibited *F. virguliforme* mycelia growth at approximately 75% [13]. These differences in growth behavior are most likely related to the recognition processes among these fungi, triggering the production of metabolites that can inhibit each other's growth. other's In fact, volatile compound-mediated recognition between *Trichoderma* spp. and *F. oxysporum* was described recently [14].

To unravel the underlying mechanisms leading to differences among the Th19A and Th4 antagonistic behavior against *F. virguliforme*, we conducted a dual-RNA sequencing experiment focusing on both *T. afroharzianum* and *F. virguliforme* transcriptomes during different stages of interaction. Our hypothesis is that significant and distinct changes in the transcriptome of *T. afroharzianum* isolates Th19A and Th4 may occur while antagonizing *F. virguliforme*, and specific changes in the *F. virguliforme* transcriptome may also occur in response to each interaction. The analysis of possible changes in *F. virguliforme* and *T. afroharzianum* transcriptomes can help identify novel candidate genes involved in fungi recognition, as well as genes that might be critical for the success of BCAs in antagonizing their prey.

## Material and methods

### Fungal isolates

*Trichoderma afroharzianum* isolates used in this study were chosen based on different antagonistic bbehaviors against *F. virguliforme* [13], characterized by overgrowth (Th19A) or formation of an inhibition zone (Th4). These isolates were obtained from a fungal collection maintained at Southern Illinois University, Carbondale, IL, and were originally isolated from soybean seedlings in the Midwest US [13]. *Fusarium virguliforme* Mont-1 (NRRL 22292) [15] was used in these experiments. Fungal isolates were grown on full-strength potato dextrose agar (PDA) before starting the experiments.

### Dual plate assay for RNA-seq

A dual plate assay using PDA-covered cellophane was carried out for the RNA-seq experiment following the methodology described by [16]. Briefly, a 6-mm diameter mycelial of *F. virguliforme* plug was transferred to a PDA plate (2.5 cm from the border) 3 days before transferring an equal-sized mycelial plug of *T. afroharzianum* to the opposite side of the same plate.

Mycelial plugs were obtained by using a sterile glass Pasteur pipette (6 mm diameter) to excise from the edges of actively growing colonies and transferred using sterile toothpicks. The plates were incubated in the dark at 25 ˚C (± 0.5 ˚C). Mycelia of *T. afroharzianum* and *F. virguliforme* were harvested for RNA extraction at three different interaction points [9]; S1 = first stage of interaction before any physical contact between the fungi (10mm apart from each other); S2 = second stage of interaction where Th19A touches Fv colony and Th4 starts formation of inhibition zone; S3 = third stage of interaction where Th19A overgrows (5mm) Fv and complete inhibition zone is formed between Fv and Th4 (S1 Fig). The cellophane interface on the PDA allowed for easy harvest of mycelia using a sterile spatula to scrape the desired growth zone. Harvested mycelia were immediately frozen in cryovials using liquid nitrogen. *Trichoderma* isolates confronting themselves, and *F. virguliforme* confronting itself were considered controls. The assay was conducted in a completely randomized design with four replicates. Each replicate consisted of six plates pooled together to have enough mycelia for RNA extraction.

## RNA isolation and sequencing

Mycelia were immediately frozen in liquid nitrogen at harvest and ground with a mortar and pestle before proceeding to RNA isolation and purification using the RNeasy mini kit and RNase-free DNase kit (Qiagen, Hilden, Germany) following the manufacturer's protocol. The isolated RNA's quality was checked using a bleach gel [17] and quantified using a Qubit 2.0 fluorometer (Life Technologies, Carlsbad, CA). Total RNA was sent to BGI, Hong Kong, China, for RNA (transcriptome) sequencing. Samples were submitted to BGI quality control to guarantee good RNA integrity (RIN value: $\geq$ 7.0 and 28S/18S: $\geq$ 2.0). Sequencing was done using a DNBSEQ-G400 platform to generate 150bp pair-end reads.

## Bioinformatic analysis of transcriptome

Clean reads from the paired-end sequencing received from BGI were uploaded to the Galaxy platform (https://usegalaxy.org/), where all bioinformatic analyses were performed. Sequence reads' quality was inspected for each sample using FastQC and MultiQC to aggregate FastQC reports [18]. Data quality control, including filtering and adapter trimming, was performed by BGI using the SOAPnuke software (). "" *Trichoderma afroharzianum* T6776 and *F. virguliforme* Mont-1 reference genomes and annotations were obtained from EnsemblFungi (http://fungi.ensembl.org/) [19, 20] and Dryad (https://datadryad.org/) [21], respectively. Clean reads from each sample were mapped to each reference genome using STAR (Spliced Transcripts Alignment to a Reference) v. 2.7.8a [22]. """" Each genome was mapped independently. The mapping results were inspected using the Integrative Genomics Viewer (IGV) [23], and the Read Distribution tool from the RSeQC tool suite [24] was used to check the distribution of reads across known gene features (exons, CDS, 5' UTR, 3' UTR, introns, intergenic regions). The quantification of the number of reads mapped per annotated gene on each reference genome was performed using featureCounts [25]. The read count files generated by featureCounts from the different samples were used to analyze the differentially expressed genes (DEG) using DESeq2 v1.12.3 [26] in R v3.6.3. (S2 Fig). The results were constructed in DESeq2 using a single factor design "design = ~treatment".

## Statistical analysis and data visualization

For data visualization, the raw count data was transformed based on the regularized logarithm (*rlog*) function using DESeq2 v1.12.3 [26] in R v3.6.3., which produces transformed data on the $\log_2$ scale that has been normalized based on library size and other normalization factors

[26]. A principal component analysis (PCA), using the *rlog* transformed counts, was used to visualize the overall effect of the treatments on the fungi transcriptomes using the plotPCA function within DESeq2. A heatmap based on *rlog* transformed counts was used to visualize gene patterns of expression of the most differentially expressed genes. Differentially expressed genes (DEGs) were determined based on contrasts between each sample and the respective control (*F. virguliforme*, Th19A, or Th4 growing alone). For further analysis, the list of DEGs for each stage of interaction was filtered based on absolute $\log_2$ fold change $> 1$ and adjusted p-value $< 0.05$ (based on FDR corrections). Venn diagrams were made to highlight uniquely expressed genes by using the InteractiVenn tool [27]. Gene ontology (GO) enrichment analysis was performed using the GOEnrichment tool V2.0.1 [28] with a p-value cut-off $< 0.01$ and Benjamini-Hochberg multiple test correction for filtered DEGs. A secreted protein profile for Th19A and Th4 was investigated based on DEGs enriched for the GO category extracellular region for each isolate's transcriptome.

## Results

### In vitro dual plate assay

The differences in the interaction behavior between Th19A and Th4 when antagonizing *F. virguliforme* were visually observed; Th19A overgrew *F. virguliforme* completely while Th4 formed an inhibition zone (Fig 1). These results have been initially observed by Warner 2016.

### Quality of sequences and mapped genes to reference genomes

Total sequences generated for each sample varied from 46,715,214 to 49,338,044 clean reads, including both forward and reverse reads. Sequences GC content varied from 53.21% to 57.37% among samples. Above 94% of the nucleotides had a quality score higher than 20 (Q20% ranging from 94.07% to 96.52%) for all samples. The number of total reads mapped to *F. virguliforme* and *T. afroharzianum* reference genomes varied across samples, with a higher number of reads mapping to the *T. afroharzianum* genome, due to the higher amount of *T. afroharzianum* tissue in the samples (S1 Table). Overall, less than 0.8% of total reads in the

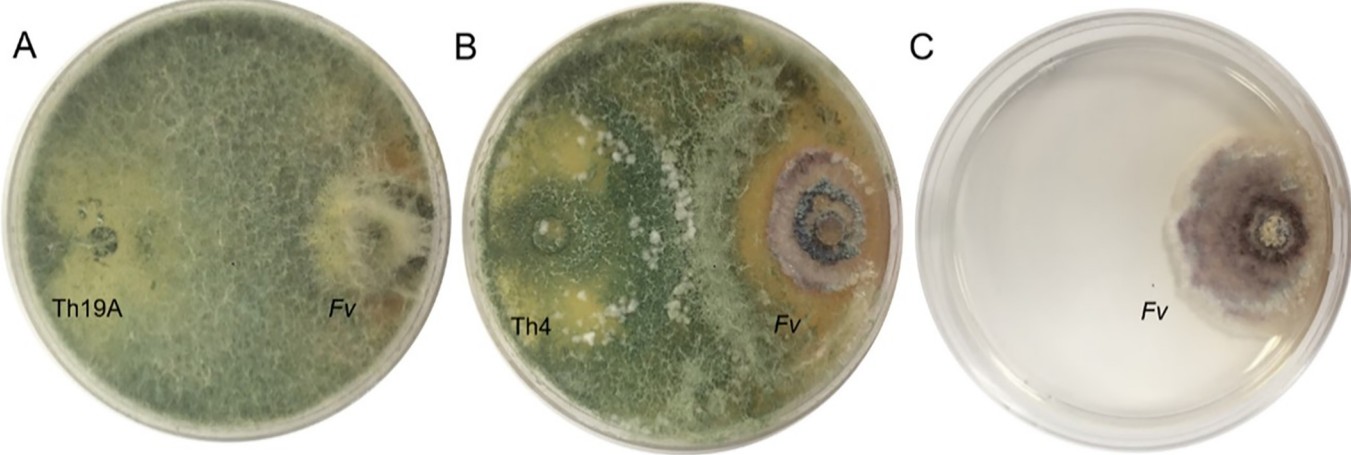

**Fig 1. Dual plate assay of two *T. afroharzianum* isolates against *F. virguliforme* (Fv) 10 days after confrontation on potato dextrose agar; A. Complete overgrowth of Fv by Th19A; B. Inhibition zone formed between Th4 and Fv; C. Control plate containing only Fv.**

control samples mapped to non-targeted genomes (*F. virguliforme* transcriptome mapping against *T. afroharzianum* genome and vice-versa) (S1 Table).

## Analysis of differentially expressed genes in *Fusarium virguliforme*

Significant changes in the *F. virguliforme* transcriptome occurred during the interaction with Th19A versus Th4, which could be visualized as separate clusters on the PCA analysis (Fig 2). Separate clusters were also observed for the different stages of the interaction, indicating that *F. virguliforme* was reacting to the antagonists' presence as the interaction advanced. Interestingly, more significant changes seemed to occur at S1 and S3, whereas *F. virguliforme* transcriptome seemed more similar to the control's at S2 (Fig 2). These changes in the *F. virguliforme* response were explored further by the analysis of DEGs, which demonstrated a similar trend across S1, S2, and S3 with both Th19A and Th4; the lowest counts for DEGs were found during S2, at the start of overgrowth, or the inhibition zone formation. Higher DEG counts were observed at S1, at the beginning of the interaction between fungi, before physical contact, followed by S3 (Fig 3 and S3 Fig). After filtering only DEGs with absolute log2 fold change > 1 and adjusted p-value <0.05, the percentage of *F. virguliforme* DEGs were

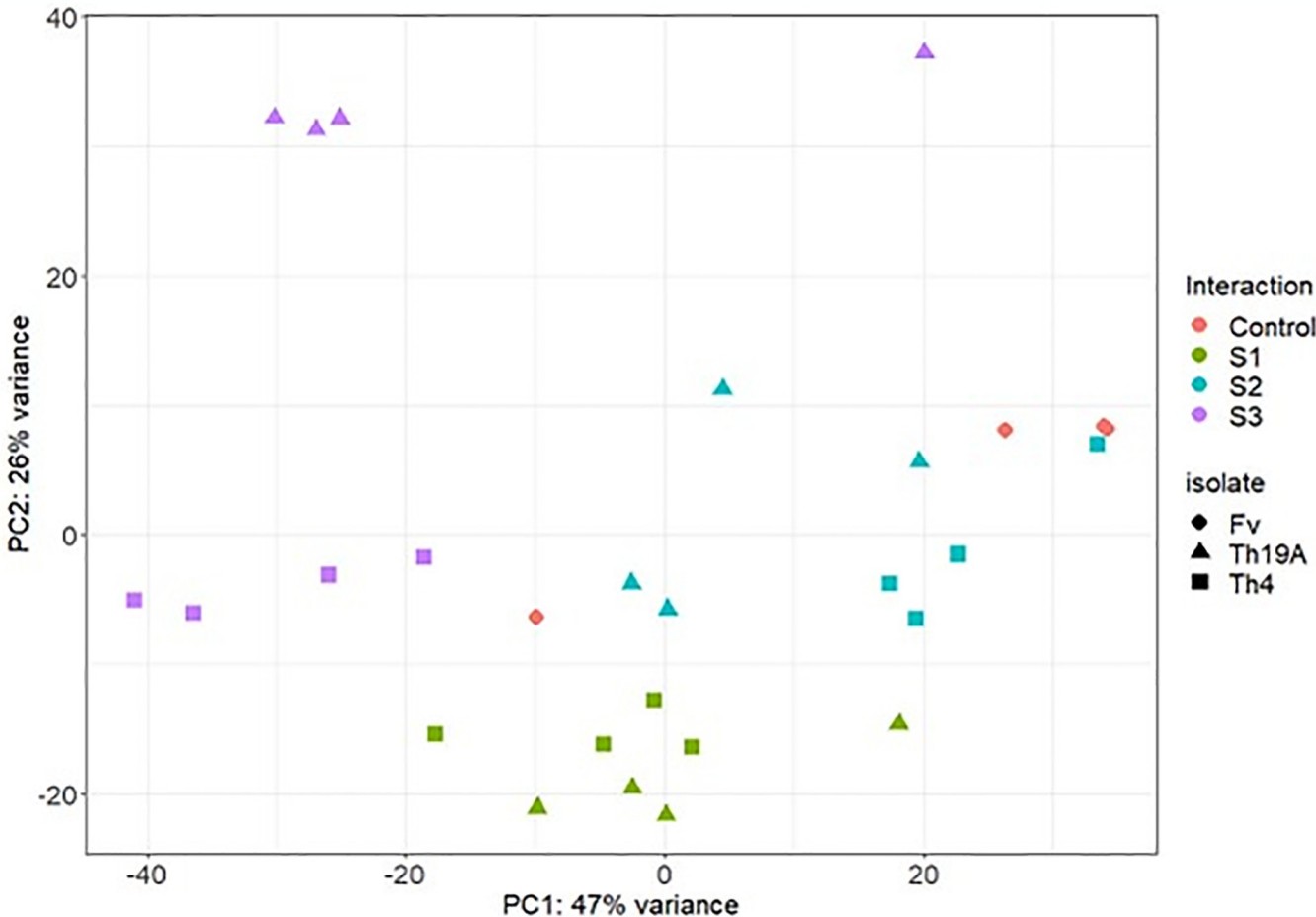

**Fig 2. Principal component analysis of the *F. virguliforme* (Fv) transcriptome at different stages of its interaction with *T. afroharzianum* isolates Th19A and Th4 in a dual plate assay.** Th19A completely overgrows Fv whereas Th4 forms an inhibition zone. Each point in the graph represents an RNA-seq sample. S1 = before physical contact, S2 = at contact or beginning of inhibition zone, S3 = overgrowth or complete inhibition zone.

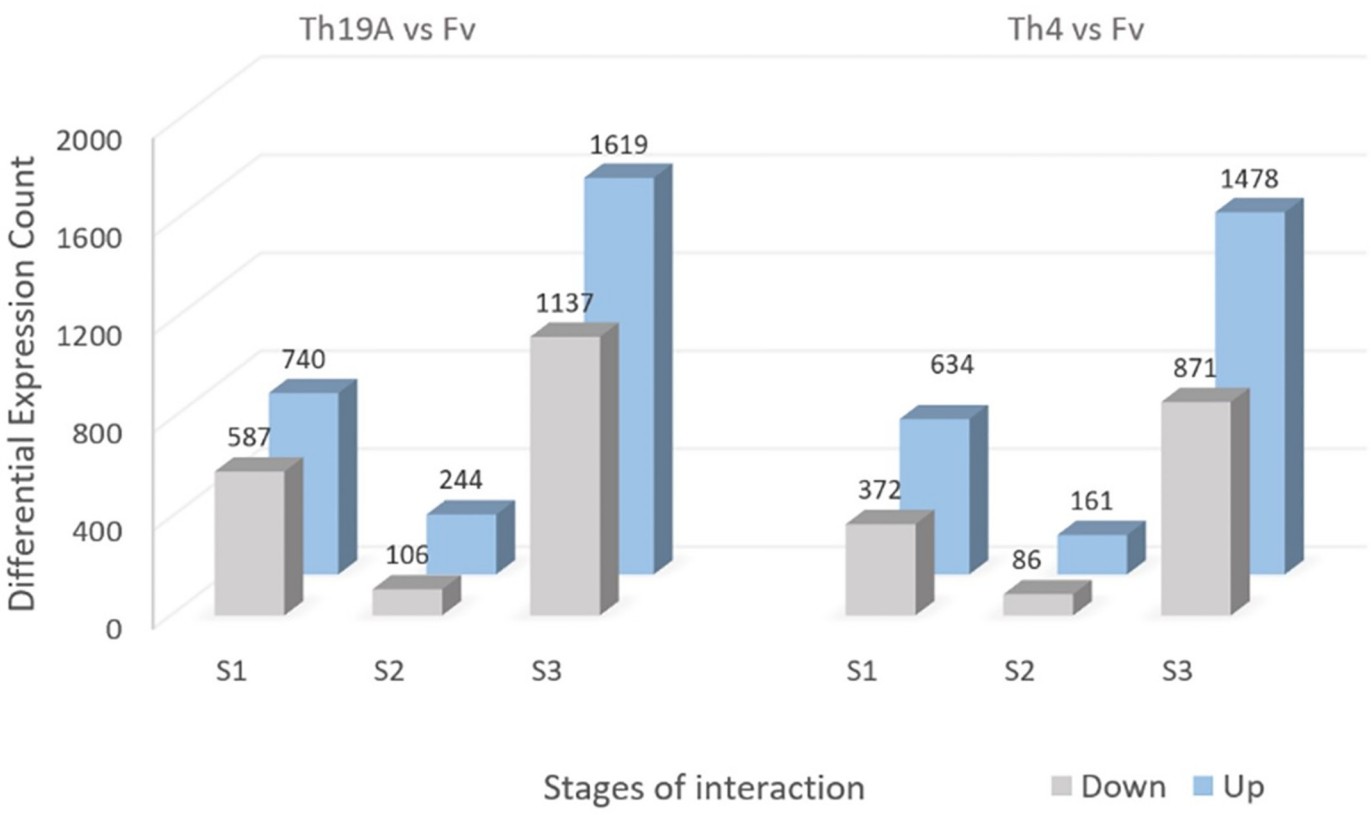

**Fig 3. *F. virguliforme* (Fv) differentially expressed genes (DEG) at different stages of the interaction with *T. afroharzianum* Th19A and Th4 during dual plate assays.** Bars represent filtered DEG counts based on adjusted p-value < 0.05 and absolute fold change > 2. DEGs for each treatment are counts contrasted with Fv growing alone.

approximately 9.5% and 7.2% at S1, 2.5% and 1.8% at S2, and 19.7% and 16.8% at S3 of the total expressed genes during interactions with Th19 and Th4, respectively (Fig 3). In addition, the amount of *F. virguliforme* up-regulated genes was higher than that of down-regulated genes during interactions with both isolates (Fig 3). Unique genes were differentially expressed during the different stages of the interaction between both fungi, as observed in the Venn diagram (Fig 4). From the 2756 and 2349 *F. virguliforme* DEGs at S3 while interacting with Th19A and Th4, respectively, about 74% (2045 genes for Th19A and 1742 genes for Th4) were uniquely expressed at S3 (Fig 4). Only 60 (17.1%) of the DEGs were uniquely expressed at S2 upon interaction with Th19A, versus 120 (48.6%) with Th4 (Fig 4).

A GO enrichment analysis was performed for the *F. virguliforme* up- and down-regulated genes. During the interaction with Th19A at S1, 71 upregulated genes were identified for the GO category catalytic activity (GO:0003824), and 22 downregulated genes for the GO category oxidoreductase (GO:0016491) (S2 Table). A total of 105 upregulated genes in both of these categories were upregulated when interacting with Th4, in addition to 18 downregulated genes within the oxidoreductase category (S2 Table). During S2, no upregulated DEGs were enriched, whereas downregulated DEGs-encoding proteins with catalytic activity and oxidoreductase activity were enriched during the interaction with Th19A, and catalytic activity only was enriched during the interaction with Th4 (S2 Table). GO enrichment analysis for *F. virguliforme* DEGs at S3 demonstrated that the majority of enriched functions are associated with genes that are being downregulated upon interaction with both BCA isolates (Fig 5).

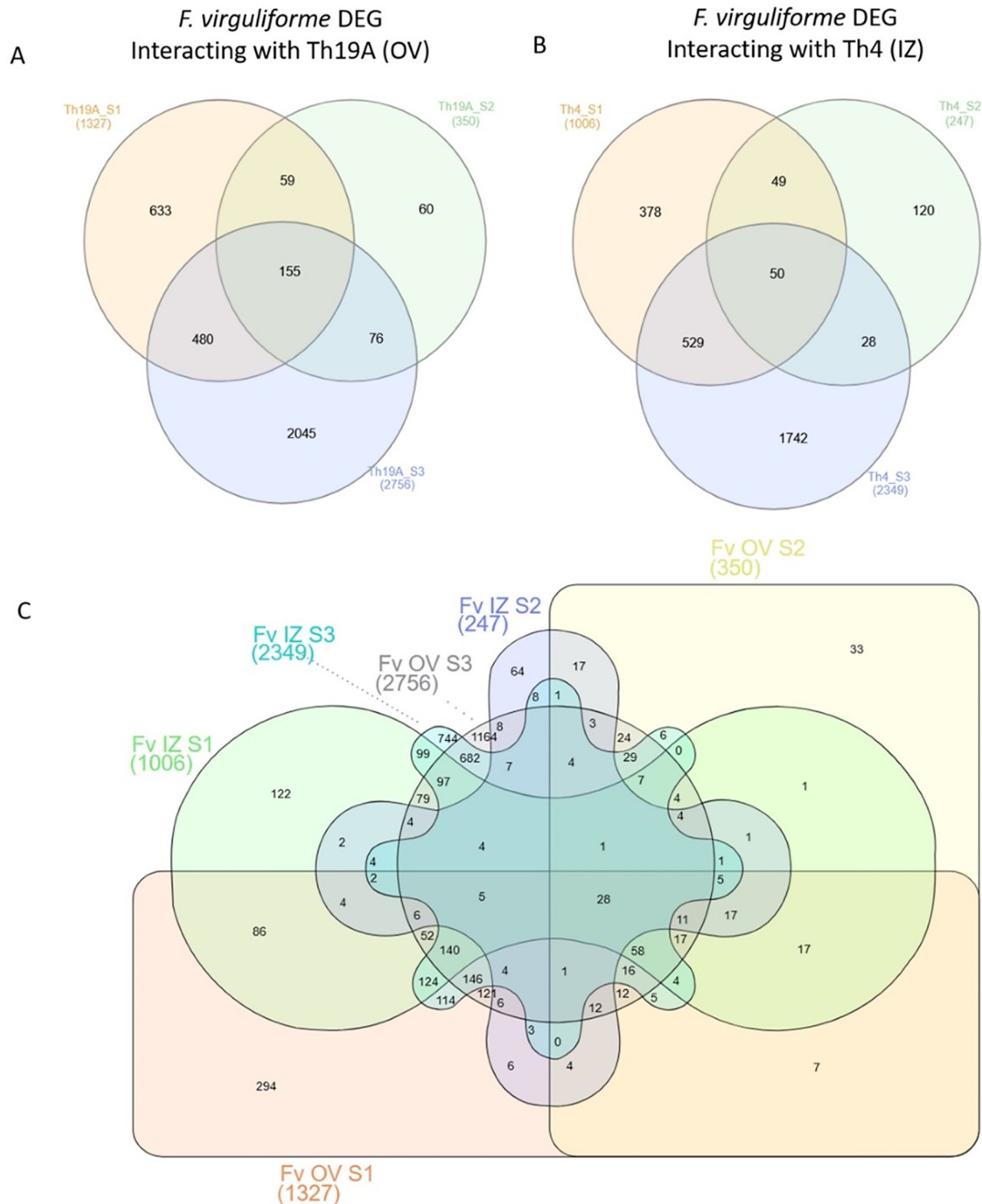

**Fig 4. Venn diagram highlighting of *F. virguliforme* (Fv) differentially expressed genes (DEG) at different stages of its interaction with *T. afroharzianum* Th19A (OV = overgrowth) and Th4 (IZ = inhibition zone).** S1 = before physical contact, S2 = at contact or beginning of inhibition zone, S3 = overgrowth or complete inhibition zone.

Most enriched GO molecular function categories associated with downregulated genes were shared whether *F. virguliforme* was overgrown or there was formation of an inhibition zone. These categories included catalytic activity, hydrolase activity (hydrolyzing O-glycosyl compounds), heme binding, and oxidoreductase activity; however, downregulated genes associated with vitamin binding (GO:0019842) were uniquely enriched upon interaction with Th19A (Fig 5), and metallopeptidase activity (GO:0008237) was uniquely enriched upon interaction with Th4 (Fig 5). GO biological process categories uniquely enriched in *F. virguliforme* down-regulated genes during overgrowth by Th19A were metabolic process (GO:0008152) and carbohydrate metabolic process (GO:0005975) (Fig 5).

Specific GO categories associated with upregulated *F. virguliforme* genes at S3 were unique for the interaction with either antagonist isolates; FAD binding (19 genes, GO:0071949) and oxidoreductase activity (41 genes, GO:0016491) were enriched during the interaction with Th19A (Fig 5). All 19 upregulated proteins containing FAD/NAD(P)-binding domains were annotated as aromatic-ring hydrolases (flavoprotein monooxygenases, Pfam PF01404, PF00890). From the 41 genes related to oxidoreductase activity, 18 were zinc-containing alcohol dehydrogenases (Pfam PF08240, PF00107, InterPro IPR020843, IPR016040), six were part of the short-chain dehydrogenase/reductase (SDR) superfamily, glucose/ribitol dehydrogenase family (InterPro IPR002347), four were polyketide synthase, an additional four were oxidoreductase family NAD(P) binding domain (Pfam PF01408), three were nitroreductase family (Pfam PF00881), and the remaining were, heme haloperoxidase family, chloroperoxidase

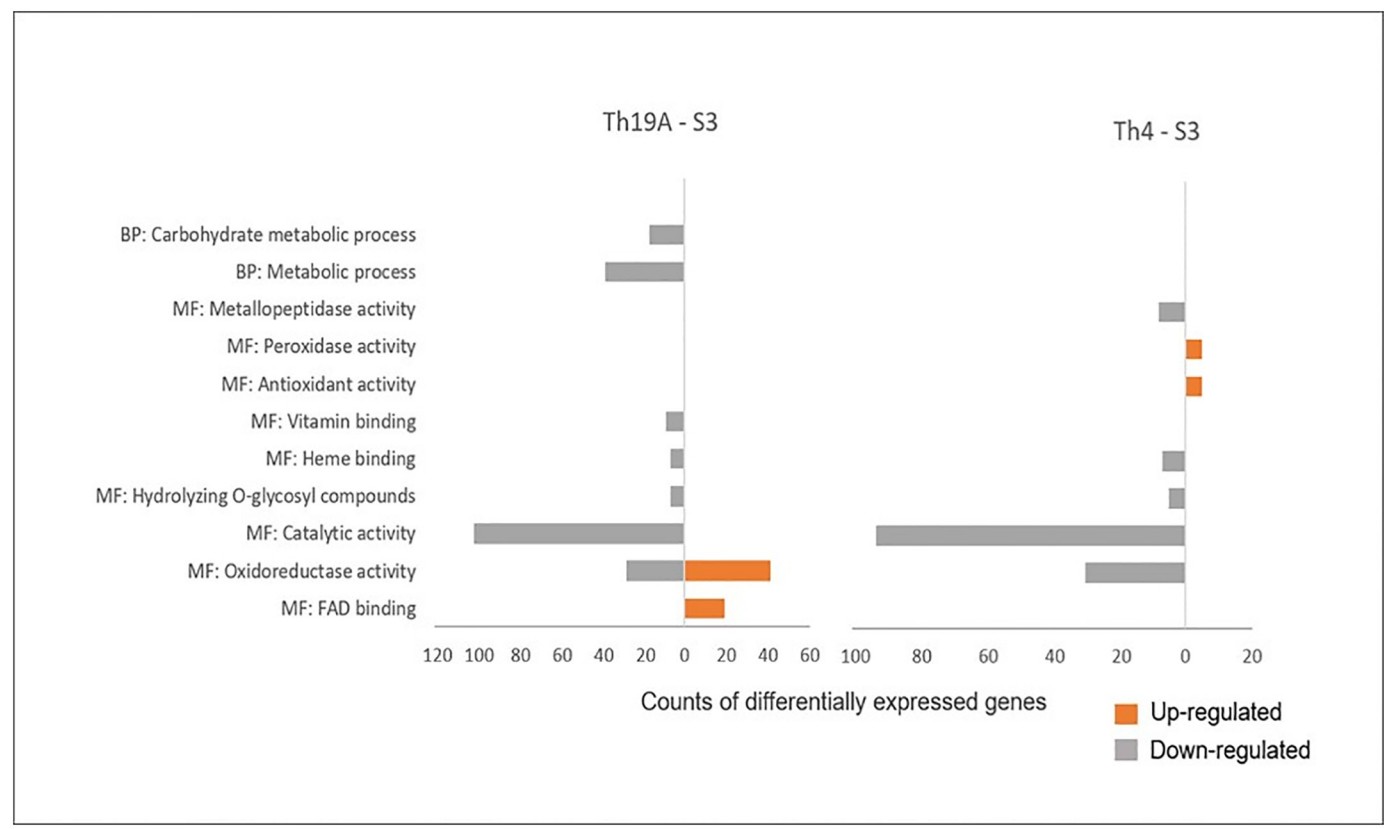

**Fig 5. Gene Ontology (GO) enrichment analysis of *F. virguliforme* differentially expressed genes upon interaction with *T. afroharzianum* Th19A (Overgowth) and Th4 (Inhibition zone) in a dual plate assay.** GOEnrichment version 2.0.1 was used for the analysis using Benjamini-Hochberg multiple test correction with a p-value cut-off <0.01.

(InterPro IPR000028), catalase (Pfam PF06628 and PF00199), cupredoxin, alcohol dehydrogenase GroEs like domain, and DSBA-like thioredoxin domain (Fig 6). Many of these specific genes, including multiple short-chain dehydrogenase/reductase (SDR) superfamily, seem to be over-expressed in *F. virguliforme* only when it is being overgrown by Th19A (Fig 6). In fact, the two upregulated *F. virguliforme* genes with the highest fold change upon interaction with Th19A at S3 were Fvm1-04074 and Fvm1-01897 ($log_2$ fold change = 14 and 13, respectively). They both belong to the short-chain dehydrogenase/reductase (SDR) superfamily, glucose/ribitol dehydrogenase family. GO categories enriched for *F. virguliforme* interaction with Th4 at S3 included antioxidant activity and peroxidase activity (5 genes, GO:0004601 and GO:0016209) (Fig 5). From these five genes, two were catalases (catalase, mono-functional, haem-containing, Pfam PF06628 and PF00199), and three were peroxidases family 2, chloroperoxidases (Pfam PF01328).

## *Fusarium virguliforme* differentially expressed genes upon interaction with Th19A and Th4 at S2

Further investigation of *F. virguliforme* transcriptome responses upon interaction with both *T. afroharzianum* isolates at S2 was performed since this stage represents the beginning of the formation of an inhibition zone with Th4 and the beginning of physical contact with Th19A. A differential gene expression analysis was conducted comparing the treatments Fv-vs-Th4 and Fv-vs-Th19A at S2. Twenty-one genes were upregulated, and 34 genes were downregulated for Fv-vs-Th4 compared to Fv-vs-Th19A (S3 Table). Three genes encoding enzymes from the major facilitator superfamily (MSF) were downregulated by approximately 3-fold; Three genes from the glucose/ribitol dehydrogenase family signature and other two short-chain dehydrogenases were strongly downregulated, ranging from 6- to 955-fold change; Additional *F. virguliforme* genes with strong downregulation during the interaction with Th4 versus Th19A encode several proteins with hydrolase, oxidoreductase, and dehydrogenase functions, as well as proteins containing different domains, including trypsin-like peptidase, methyltransferase, rhodanese-like domain, NAD(P)-binding domain, fungal Zn(II)-Cys(6) binuclear factor (fungal transcription factor), glutathione S-transferase N-terminal, and aconitase C-terminal domain (S3 Table).

The most upregulated genes in *F. virguliforme* during the interaction with Th4 in comparison to Th19A at S2 encode for glycosyl hydrolase (GH family 45), a protein from the phosphoesterase family, and a zinc-binding dehydrogenase, which expression was increased by 12-, 10-, and 9-fold change, respectively (S3 Table). Interestingly, a gene encoding a peptidase inhibitor from the subtilase family, and an amino acid permease was upregulated by a 3-fold change. Additional upregulated genes are included in the aldehyde dehydrogenase family, GH family 45, MSF, and phosphoesterase family (S3 Table). Four upregulated genes have no known function, and other two genes are proteins predicted to be embedded in the membrane. The set of up and downregulated genes described here may play an essential role in the formation of the inhibition zone between *F. virguliforme* and Th4.

## Analysis of *Trichoderma afroharzianum* differentially expressed genes

Significant changes in Th19A and Th4 transcriptomes were visible in the different stages of interaction with *F. virguliforme*, as highlighted by the tight clusters observed in the PCA analysis (Fig 7). At S1, Th19A and Th4 transcriptomes were similar to the respective control, where each isolate was growing without the presence of *F. virguliforme* (Fig 8 and S4 Fig). Both isolates seem to have differences in terms of their transcriptome, even without the presence of *F. virguliforme* (Fig 7). During S2 and S3, the responses of Th19A and Th4 to the interaction with

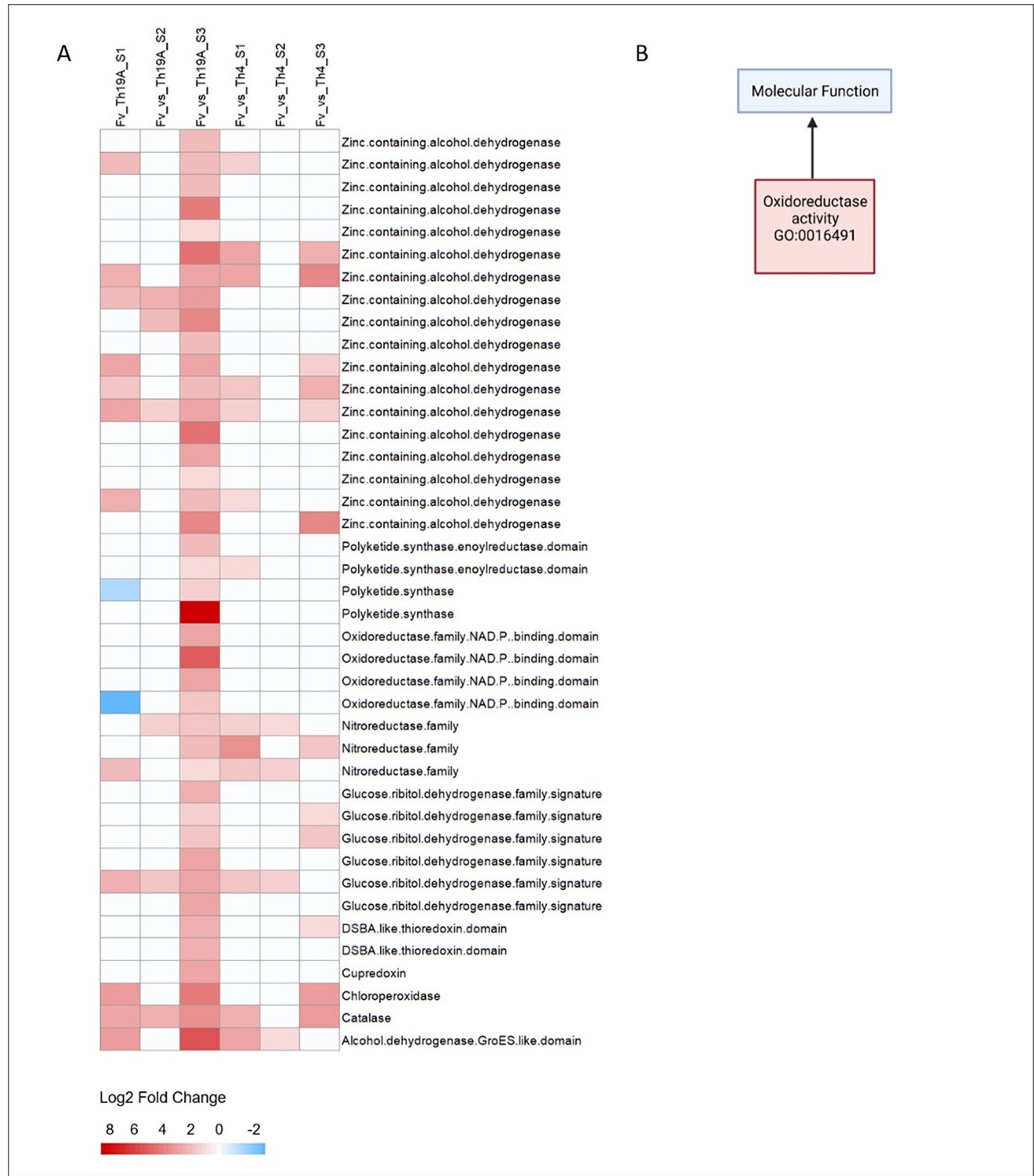

**Fig 6. Heatmap of the Log2 fold change of *F. virguliforme* (Fv) differentially expressed genes.** (A) Forty-one up-regulated Fv genes enriched for the GO category oxidoreductase activity (GO:0016491) while being overgrown by Th19A (S3) and their log$_2$ fold change for the additional interactions; (B) GO enriched category.

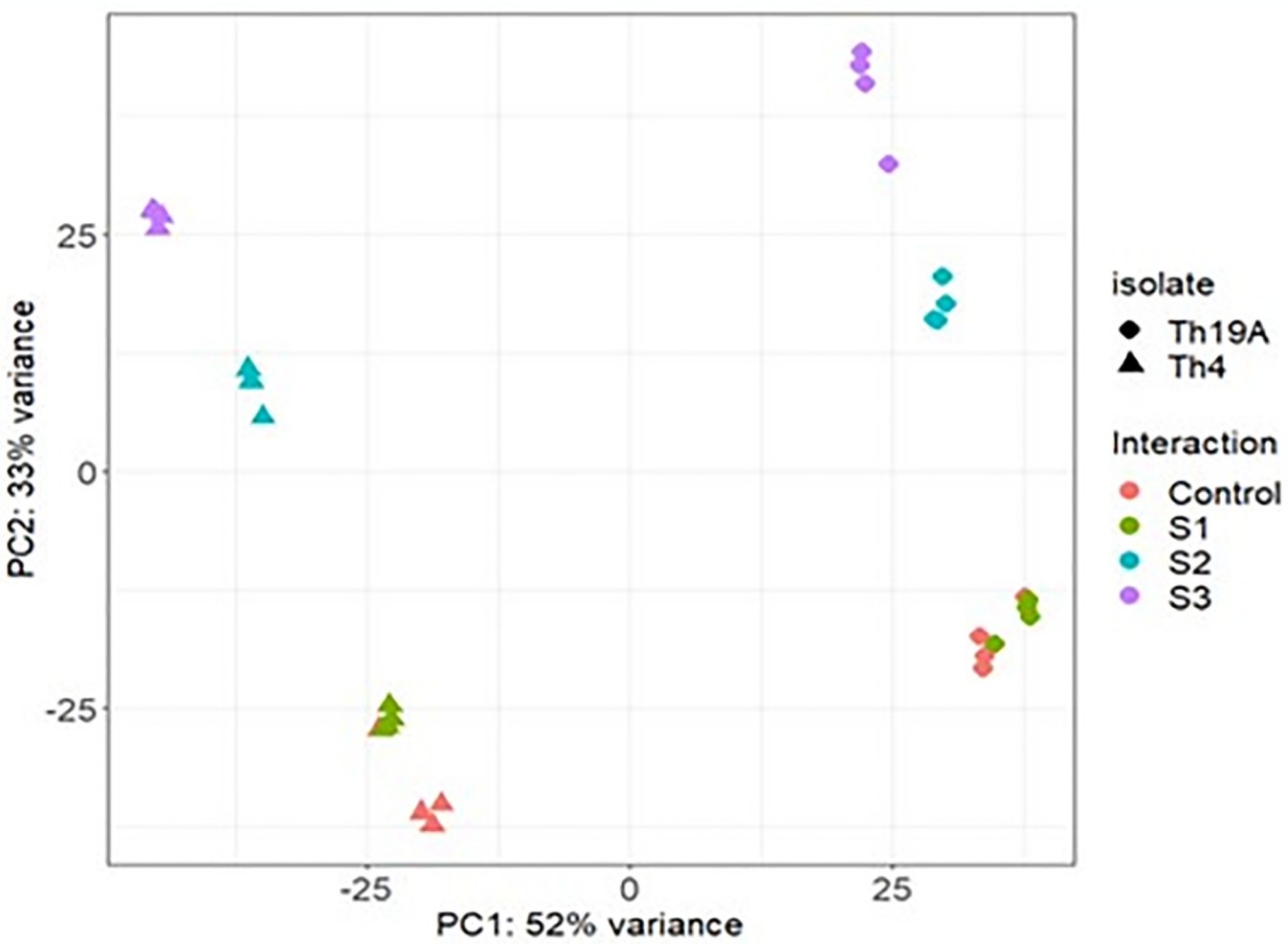

**Fig 7. Principal component analysis of *T. afroharzianum* Th19A and Th4 transcriptomes during the different stages of interaction with *F. virguliforme* (Fv) in a dual plate assay.** Th19A completely overgrows Fv whereas Th4 forms an inhibition zone. Each point in the graph represents an RNA-seq sample. S1 = before physical contact, S2 = at contact (or beginning of inhibition zone formation, S3 = overgrowth or complete inhibition zone.

the pathogen were stronger, with an increasing number of DEGs (Fig 8). The number of up-regulated genes was higher in all stages of interaction for Th19A compared to down-regulated genes, whereas, for Th4, the number of down-regulated genes was higher at S1 and S2 compared to the up-regulated genes (Fig 8). In contrast to *F. virguliforme* response where DEG was higher at S3 and S1 and lowest at S2, *T. afroharzianum* had increasing counts for DEG as the interaction progressed for both isolates (Fig 8). Unique genes are differentially expressed at the different stages of the interaction between both fungi (Fig 9).

GO enrichment analysis on *T. afroharzianum* DEGs demonstrated that some functions were uniquely enriched for the different isolates during the interaction with *F. virguliforme* (Fig 10). During S1, upregulated DEGs assigned to catalytic activity and polysaccharide catabolic process were enriched for both BCA isolates. However, additional functions were enriched for Th19A, including oxidoreductase activity (GO:0016491), metabolic process (GO:0008152), and membrane (GO:0016020) (Fig 10). Downregulated DEGs enriched for cellular anatomical entity (GO:0110165) was unique for Th4 at S1 (Fig 10). During S2, additional

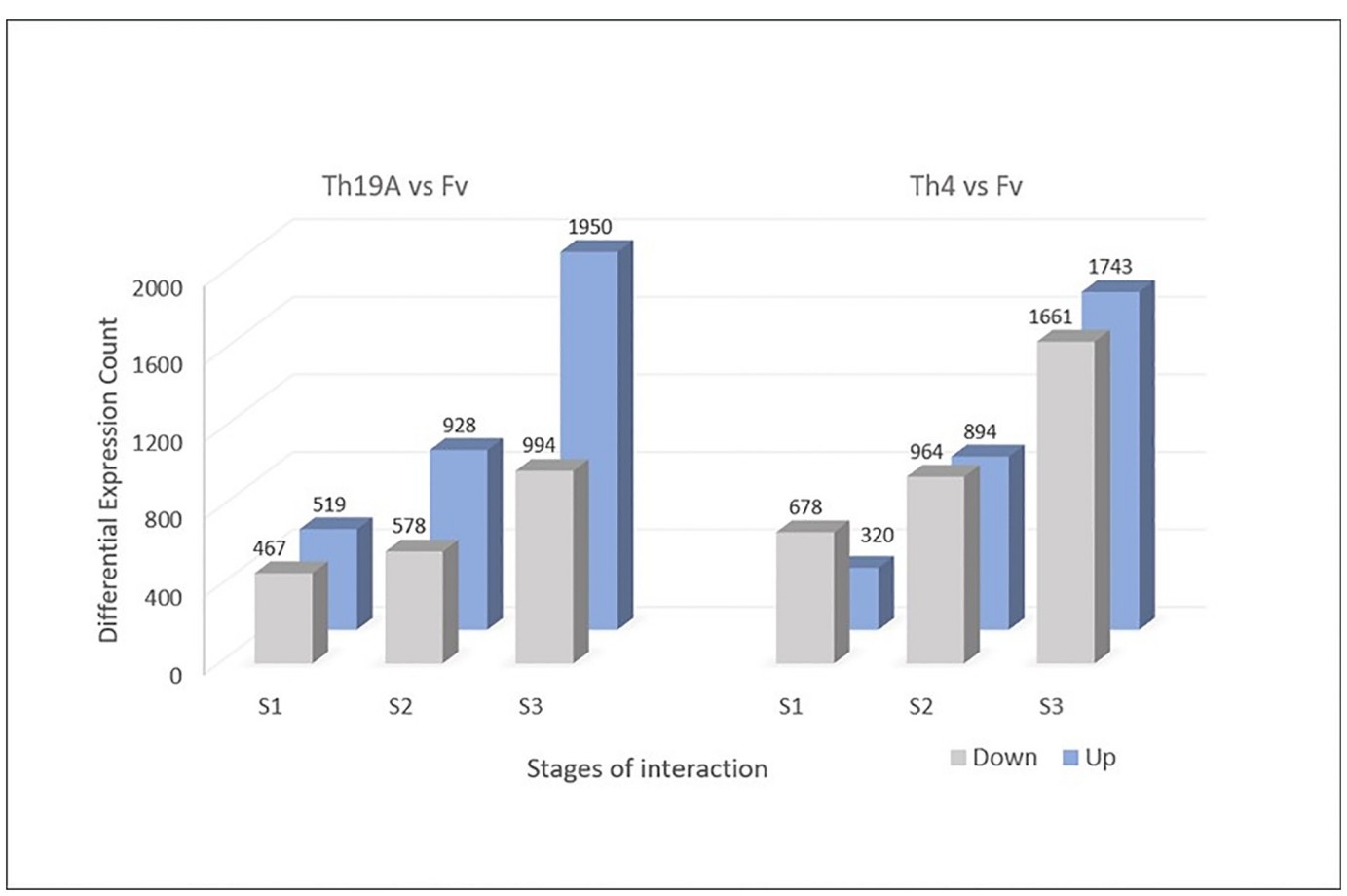

**Fig 8. Differentially expressed genes (DEG) of *T. afroharzianum* Th19A and Th4 at different stages of the interaction with *F. virguliforme* in dual plate assays.** Bars represent filtered DEG counts based on adjusted p-value < 0.05 and absolute fold change >1. DEG for each treatment are counts contrasted with Th19A and Th4 growing alone.

functions were uniquely associated with Th19A upregulated DEGs, including anion binding (GO:0043168), 3-oxoacyl-[acyl-carrier-protein] synthase activity (GO:0004315), carbohydrate metabolic process (GO: 0005975), and metabolic process (GO:0008152).

Several GO categories were uniquely enriched in the Th4 downregulated DEGs at S2, including endopeptidase activity (GO:0004175), peptidase activity (GO:0008233), and serine-type endopeptidase activity (GO:0004252) (Fig 10). GO enrichment categories shared between Th19A and Th4 up and down-regulated DEGs at S3 included catalytic activity, oxidoreductase activity, membrane, and cellular anatomical entity. Notice that upregulated DEGs enriched for oxidoreductase activity were abundant in the Th9A transcriptome from the beginning of the interaction with *F. virguliforme* (Fig 10A), whereas they were only upregulated in the Th4 transcriptome at S3 (Fig 10B). An inverse trend is observed for the DEGs enriched for cellular anatomical entity, being downregulated consistently in Th4 during all interaction times, but only downregulated in Th9A at S3 (Fig 10). Additional enriched GO categories at S3 unique to Th19A upregulated DEGs were hydrolase activity, hydrolyzing O-glycosyl compounds (GO:0004553), extracellular region, polysaccharide catabolic process (GO:0000272), and metabolic process. It is to be noted that DEGs enriched for extracellular region were upregulated earlier in the Th4 transcriptome (at S2) compared to Th9A (at S3) (Fig 10). Furthemore, DEGs

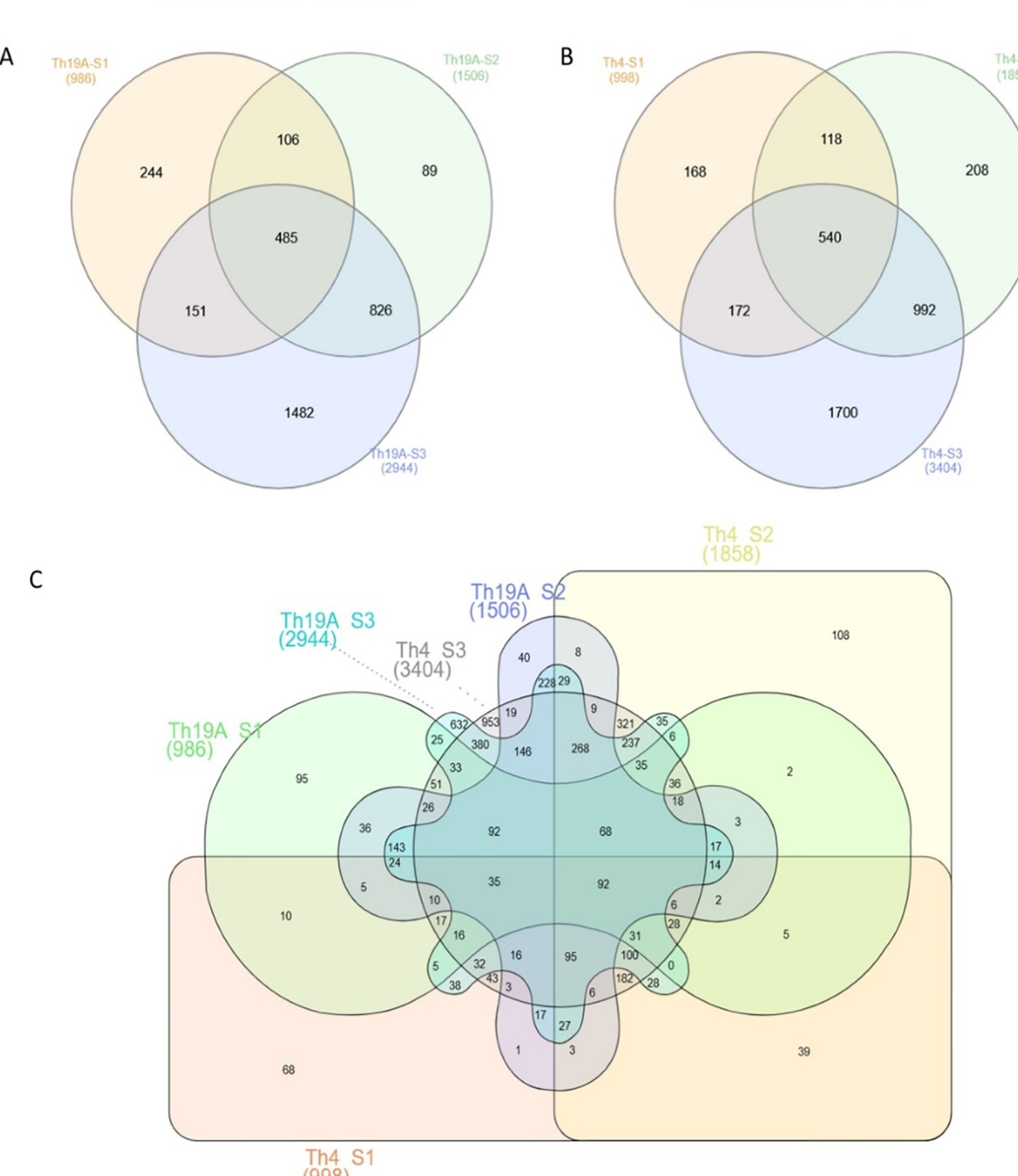

**Fig 9. Venn diagram highlighting differentially expressed genes (DEG) of *T. afroharzianum* Th19A (overgrowth) and Th4 (inhibition zone) at different stages of the interaction with *F. virguliforme*.** S1 = before physical contact, S2 = at contact or beginning of inhibition zone, S3 = overgrowth or complete inhibition zone.

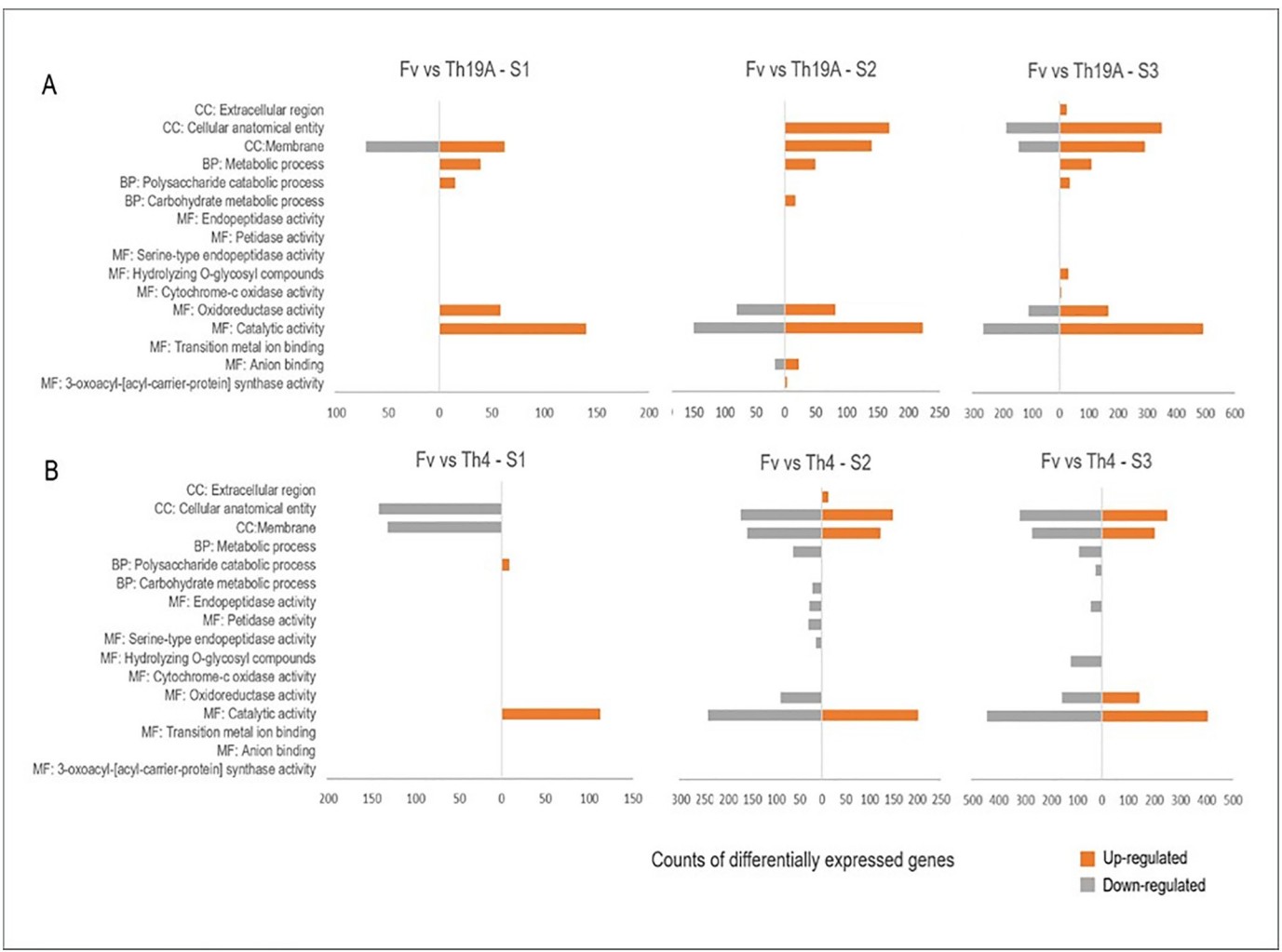

**Fig 10. Gene Ontology (GO) enrichment analysis of the differentially expressed genes (DEGs) of *T. afroharzianum* Th19A (A) and Th4 (B) during the interaction with *F. virguliforme* at different stages of the interaction.** GOEnrichment version 2.0.1 was used for the analysis using Benjamini-Hochberg multiple test correction with a p-value cut-off <0.01.

categorized within metabolic process, polysaccharide catabolic process (GO:0000272), and hydrolyzing O-glycosyl compounds, were upregulated in Th9A, while these same categories were downregulated in Th4 at S3. Cytochrome-c oxidase activity (GO:0004129) was only upregulated in Th9A at S3, when this isolate is overgrowing *F. virguliforme*. DEGs enriched for metabolic process (GO:0008152) were another interesting point of contrast, as they were uniquely upregulated in Th9A (S1, S2, and S3) but downregulated in Th4. Interestingly, endopeptidase activity was repressed in Th4 at S2 and S3 during the interaction with *F. virguliforme*. Overall, DEGs within more GO categories seem to be upregulated in Th9A and downregulated in Th4 (Fig 10).

### Secreted proteins profile differs between Th19a and Th4 upon interaction with *F. virguliforme*

Further investigation of the genes enriched within the GO category extracellular region revealed several enzymes that may play an important role in the interaction between *T.*

*afroharzianum* and *F. virguliforme*. Eight out of twenty-seven proteins were CFEM domain-containing protein (RBT5 family), and they demonstrated a variable profile of expression, some being upregulated only with Th19A, Th4, or both (Fig 11). Multiple genes encoding carbohydrate-active enzymes (CAZymes) were also upregulated during the interaction with *F. virguliforme*, with higher expression observed at S3. CAZyme encoding genes were cutinase

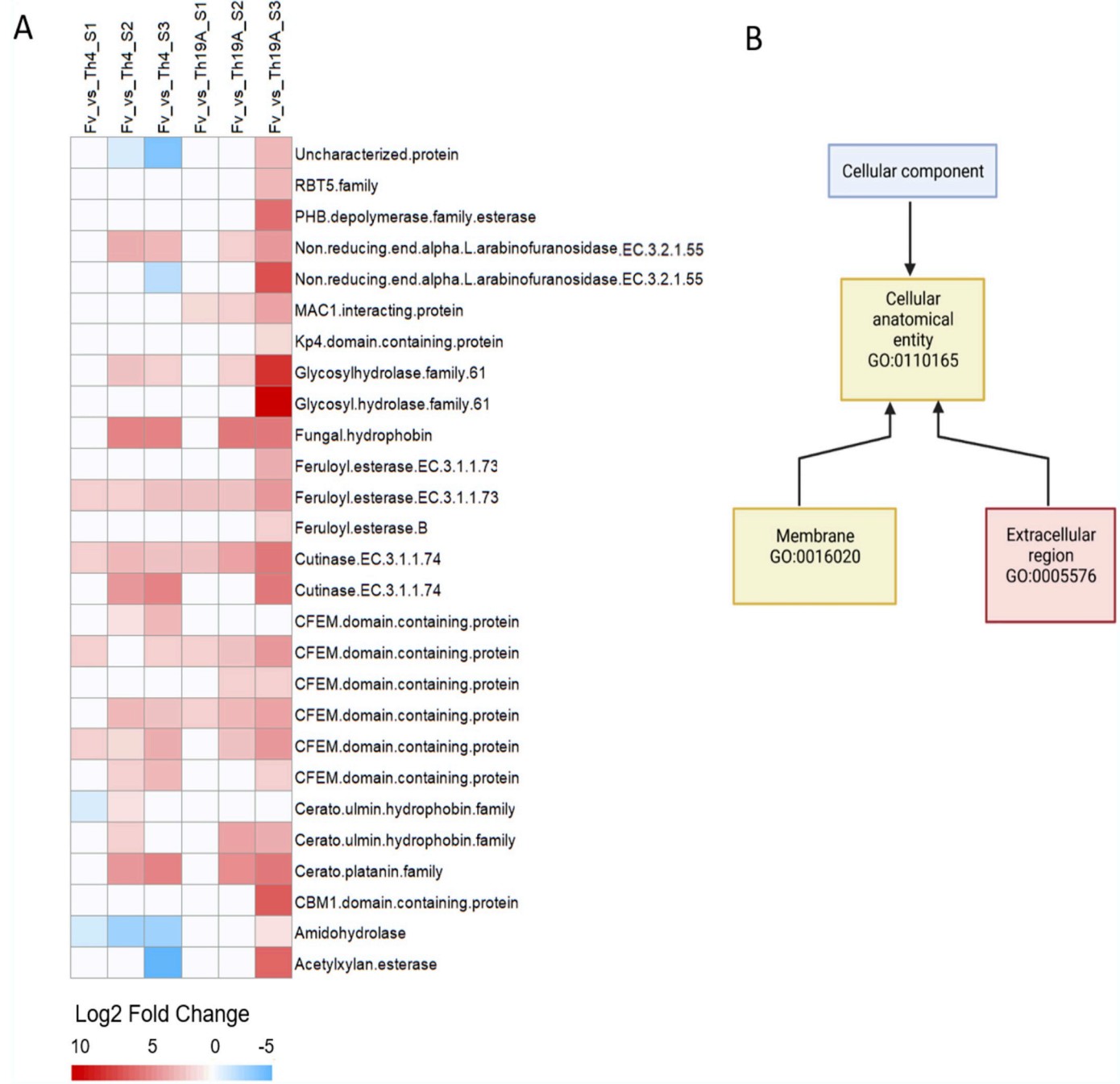

**Fig 11. Heatmap of the Log2 fold change of *T. afroharzianum* differentially expressed genes (A) annotated to GO cellular component, which were enriched for the category extracellular region (GO:0005576); (B) during enrichment analysis.**

and acetyl xylan esterase (carbohydrate esterase family 5), feruloyl esterase, non-reducing end alpha-L-arabinofuranosidase (xyloglucan degradation, GH43), and glycosyl hydrolase family 61 (copper-dependent lytic polysaccharide monooxygenases, reclassified in family AA9). The expression of some CAZymes was stronger in Th19A compared to Th4. For example, one of the non-reducing end alpha-L-arabinofuranosidase enzymes was upregulated in Th19A at S3 but downregulated in Th4 (Fig 11). Two carbohydrate-binding module (CBM1) enzymes within the extracellular GO category, namely CBM1 domain-containing protein and PHB depolymerase family esterase, were only upregulated in Th19 while overgrowing *F. virguliforme*, with no differential expression in Th4 (Fig 11). Similarly, a killer toxin Kp4 domain-containing protein was also upregulated only in Th19A at S3 while overgrowing *F. virguliforme*, with no differential expression in Th4. Moreover, an uncharacterized protein was upregulated in Th19A at S3, while being downregulated in Th4 at S2 and S3 (Fig 11). This uncharacterized protein has CBM1, expansin-like CBD, and expansin-like EG45 domains (THAR02_08479, KKO99424). Three proteins from the cerato-ulmin hydrophobin family (including fungal hydrophobin) and one protein from the cerato-platanin protein (CPP) family were upregulated at S2 and S3 in both *T. afroharzianum* isolates, but no significant expression was observed atS1 (Fig 11). A gene encoding an amidrohydrolase (peptidase M20A family) had different expression profiles between both *T. afroharzianum* isolates (Fig 11); it was highly upregulated in Th19A only while overgrowing *F. virguliforme*. However, the gene-encoding amidrohydrolase was downregulated in Th4 for all three stages of the interaction with the pathogen.

## Discussion

The present study contributes to a better understanding of the biology and molecular mechanisms governing the *F. virguliforme-T. afroharzianum* interaction. Differences in *F. virguliforme* and *T. afroharzianum* transcriptomes have been described, highlighting transcriptomic changes among *T. afroharzianum* isolates expressing a different phenotype upon interaction with the pathogen. This knowledge has important applications for the development and improvement of biocontrol tools that can be used to help manage *F. virguliforme* and other soil-borne pathogens.

Fungal proteins and genes associated with biocontrol mechanisms can be divided into five categories [29], host recognition and genetic reprogramming; activation of biosynthesis pathways of secondary metabolites (antibiosis); competition for nutrients and root colonization; mycoparasitism and secretion of cell wall-degrading enzymes; and induction of plant defenses [29]. Notwithstanding, mutual antagonistic-host recognition can lead to a chemical warfare, where the release of volatiles compounds and secreted bioactive molecules will modulate the outcome of the interaction, for example, a complete overgrowth of the host by the antagonist or formation of an inhibition zone [6, 14]. Mutual recognition of *Trichoderma* spp. and *F. oxysporum* isolates was documented by [14], where *Trichoderma* spp. significantly increased the amount/activity of secreted antifungal metabolites in response to volatile compounds (VCs) produced by *F. oxysporum*. In the present study, one of the *F. virguliforme* responses during the complete formation of an inhibition zone with Th4 was to increase the production of catalases and chloroperoxidases [27]. Catalases are common enzymes found in nearly all living organisms, with the function of protecting the cells against oxidative damage caused by reactive oxygen species (ROS); Chloroperoxidase (CPO) is a versatile heme-containing enzyme that exhibits peroxidase, catalase, cytochrome P450-like activities, and perform halogenation reactions [30]. These results indicate that, possibly, *F. virguliforme* displays a robust protective oxidative stress response against Th4 during mutual growth inhibition. NADPH oxidase

(Nox) genes are involved in the production of ROS during antagonistic interactions within fungi; ROS production has been reported as a mechanism of defense against other fungi. In *T. harzianum*, the overexpression of *nox1* lead to increased ROS production during confrontation with *Pythium ultimum*, and consequently enhanced antagonistic activity against the pathogen [31]. On the other hand, *F. virguliforme* response while being overgrown by Th19A was to increase the production of proteins related to FAD binding and several oxidoreductases (Fig 5). The downregulation of *F. virguliforme* genes related to metabolic process and carbohydrate metabolic process might be a response to the mycoparasitism by Th19A.

During the S2 stage of interaction, several genes encoding enzymes were significantly downregulated in *F. virguliforme* while interacting with Th4 in comparison to Th19A (S3 Table). Two of these enzymes belong to the Zn(II)-Cys(6) binuclear cluster domain, also called zinc cluster proteins. These proteins are exclusively found in fungi, and are associated with a multitude of important functions for fungi development and pathogenesis, including melanin biosynthesis in *Colletotrichum lagenarium* and *Magnaporthe grisea*, switching between biotrophy and necrotrophy during infection in *C. lindemuthianum*, regulation of nitrate assimilation in *Aspergillus nidulans*, activation of several thiamine-repressible genes in *Schizosaccharomyces pombe*, and the involvement in the regulation of fumonisin biosynthesis in *F. verticillioides*, among others [32]. Moreover, compared to the interaction with Th19A, three major facilitator superfamily (MFS) proteins were downregulated in *F. virguliforme* by approximately 3- fold when interacting with Th4 at S2. MSF are transmembrane transporters important to prevent intracellular accumulation of secondary metabolites during biosynthesis and play an important role in virulence and resistance to antifungal agents [33]. Interestingly, upregulated *F. virguliforme* genes upon interaction with Th4 compared to Th19A included glycosyl hydrolases (GH12 and GH45) and a peptidase inhibitor. These enzymes may play an important role in the formation of the inhibition zone.

Focusing on differences in the *T. afroharzianum* transcriptome, we observed the expression of genes related to lytic enzymes, antifungal secondary metabolites, signal transduction related genes, and cell wall hydrolases. These genes are key for the antagonism of *Trichoderma* but can be host-specific and expressed at specific times during the interaction with host fungi [34]. In this study, Th19A up- regulated hydrolases, polysaccharide catabolic process, and metabolic process while overgrowing *F. virguliforme*, whereas Th4 downregulates those same functions when forming an inhibition zone. The investigation of *T. afroharzianum* genes enriched for the GO category extracellular region revealed expression profiles of gene-encoding secreted enzymes that may play a key role in the *T. afroharzianum-F. virguliforme* interaction. Eight of these genes encode for CFEM domain-containing protein (RBT5 family), which had a variable pattern of expression between the isolates and stages of interaction. CFEM domain is a fungal specific cysteine rich domain found in some proteins with proposed roles as effector proteins [35]. Multiple *T. afroharzianum* genes encoding CAZymes were upregulated upon interaction with *F. virguliforme*. CAZymes produced by biocontrol agents play a major role in the breakdown of the fungal target's cell wall and are key enzymes enabling mycoparasitic interactions [36, 37]. In this study, a higher expression of glycosyl hydrolases was observed in Th19A while overgrowing *F. virguliforme*, which is expected due to this isolate's proven ability to mycoparasitize *F. virguliforme*, as observed by [38]. The upregulation of some of these genes occurred even before physical contact, indicating that fungal recognition happens very early in the interaction. Similar results were described by [9], where *T. atroviridae*, and *T. virens* exhibited different transcriptomic responses to *Rhizoctonia solani* before physical contact. Th4 also upregulated the expression of glycosyl hydrolases even though physical contact with the pathogen was limited by the inhibition zone.

Two *T. afroharzianum* CBM1 enzymes, CBM1 domain-containing protein and PHB depolymerase family esterase, and a killer toxin Kp4 domain-containing protein were only upregulated at S3 in Th19A, but not in Th4 or at earlier stages of interaction (Fig 11). CBMs are appended to carbohydrate active enzymes that degrade insoluble polysaccharides [39], whereas Kp4 is a family of killer toxins (PF09044) secreted by some fungal species targeting sensitive cells from the same or related fungal species, often functioning by creating pores in the target cell membrane. These enzymes may play an important role in *T. afroharzianum* antagonistic activity against *F. virguliforme*. Three proteins from the cerato-ulmin hydrophobin family (which includes fungal hydrophobin) were upregulated at S2 and S3 in both *T. afroharzianum* isolates. Fungal hydrophobins are small extracellular proteins, generally found on the outer surface of conidia and hyphal walls and are thought to be involved in mediating communication and contact between the fungus and its environment [40]. Similarly, CPP family was also found to be upregulated for both Th19a and Th4 at S2 and S3, but not at S1. CPPs are only found in filamentous fungi and are shown to play an important role during the interaction of fungi and other organisms [41]. They are abundantly secreted into culture media, but they can also be bound in the fungal cell wall [42]. CPPs act as virulence factors for phytopathogens or as elicitors of plant defenses in the case of plant beneficial biocontrol fungi. CPPs were found to affect *T. harzianum* self-recognition, as well as regulate mycoparasitism-related gene expression, modulate hyphal coiling, and regulate plant defenses [43]. Another *T. afroharzianum* gene with the extracellular category that showed different expression patterns among isolates encodes an amidrohydrolase (peptidase M20A family). This gene was upregulated in Th19A only while overgrowing *F. virguliforme*. However, it was downregulated in Th4 during all three stages of the interaction with the pathogen. Overall, the upregulation of genes coding for proteins associated with cell membrane and extracellular region for both Th19A and Th4 at S3 possibly indicates increased activity of secreted antifungal metabolites in response to the recognition of *F. virguliforme*. This study provided foundational knowledge for a deeper understanding of the molecular mechanisms underlying the antagonist-pathogen interactions in *T. afroharzianum-F. virguliforme* system.

## Supporting information

**S1 Fig. Overview of the RNA-seq analysis to evaluate the antagonistic interaction between *T. afroharzianum* and *F. virguliforme*.**
(JPG)

**S2 Fig. Scheme of the dual plate assay showing different interactions of *F. virguliforme* (Fv) with the *T. afroharzianum* isolates Th19A and Th4.** Red dotted areas represent areas where fungal mycelia were collected for RNA isolation and sequencing. S1 = first stage of interaction before any physical contact between fungi; S2 = second stage of interaction where Th19A touches Fv colony and Th4 starts formation of inhibition zone; S3 = third stage of interaction where Th19A overgrows Fv and complete inhibition zone is formed between Fv and Th4.
(JPG)

**S3 Fig. MA-plots generated based on *F. virguliforme* (Fv) transcriptome changes upon interacting with *T. afroharzianum* Th19A (A,B,C) and Th4 (D,E,F).** MA-plots illustrate the log fold change of the Fv differentially expressed genes (red). S1 = before physical contact, S2 = at contact (Th19A) or beginning of inhibition zone (Th4), S3 = overgrowth (Th19A) or complete inhibition zone (Th4). Differentially expressed genes were determined using DESeq2

and normalized against the control (Fv growing alone).
(JPG)

**S4 Fig. MA-plots generated based on *T. afroharzianum* Th19A (A,B,C) and Th4 (D,E,F) transcriptomes during the three stages of interaction with *F. virguliforme* (Fv).** MA-plots illustrate the log fold change of the Th19A and Th4 differentially expressed genes (DEGs) in red. S1 = before physical contact, S2 = at contact for or beginning of inhibition zone formation, S3 = overgrowth or complete inhibition zone. DEGs were determined using DESeq2 (False discovery rate (FDR) < 0.1) and normalized against the control (Th19A and Th4 growing alone).
(JPG)

**S1 Table. The sum of uniquely mapped reads aligned to the *F. virguliforme* and *T. afroharzianum* reference genomes at each treatment and stage of interaction.**
(XLSX)

**S2 Table. Gene Ontology (GO) enrichment analysis of *F. virguliforme* differentially expressed genes upon interaction with Th4 at S1 and S2.**
(XLSX)

**S3 Table. Differentially expressed genes (DEG) of *F. virguliforme* upon interaction with Th4 over Th19A at S2 (DESeq contrast Fv-vs-Th4 over Fv-vs-Th19A).**
(XLSX)

## Author Contributions

**Conceptualization:** Mirian F. Pimentel, Jason P. Bond, Ahmad M. Fakhoury.

**Data curation:** Mirian F. Pimentel, Leonardo F. Rocha.

**Formal analysis:** Mirian F. Pimentel, Arjun Subedi.

**Funding acquisition:** Jason P. Bond, Ahmad M. Fakhoury.

**Investigation:** Mirian F. Pimentel, Arjun Subedi, Jason P. Bond.

**Methodology:** Mirian F. Pimentel, Arjun Subedi, Ahmad M. Fakhoury.

**Project administration:** Mirian F. Pimentel.

**Resources:** Ahmad M. Fakhoury.

**Software:** Mirian F. Pimentel, Leonardo F. Rocha.

**Supervision:** Jason P. Bond, Ahmad M. Fakhoury.

**Validation:** Leonardo F. Rocha, Jason P. Bond, Ahmad M. Fakhoury.

**Visualization:** Mirian F. Pimentel.

**Writing – original draft:** Mirian F. Pimentel, Ahmad M. Fakhoury.

**Writing – review & editing:** Mirian F. Pimentel, Leonardo F. Rocha.

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
