## [Decision Letter · Decision Letter 0]

16 Apr 2024

PONE-D-24-00993Dual RNA-Seq reveals transcriptome changes during Fusarium virguliforme -Trichoderma afroharzianum  interactionsPLOS ONE

Dear Dr. Rocha,

Thank you for submitting your manuscript to PLOS ONE. After careful consideration, we feel that it has merit to be considered fr publication in  PLOS ONE’s publication criteria as it currently stands. Therefore, we invite you to submit a revised version of the manuscript that addresses the points raised during the review process.

We look forward to receiving your revised manuscript.

Kind regards,

Vivek Sharma, PhD

Academic Editor

PLOS ONE

Journal Requirements:

3. We notice that your supplementary [figures/tables] are included in the manuscript file. Please remove them and upload them with the file type 'Supporting Information'. Please ensure that each Supporting Information file has a legend listed in the manuscript after the references list.

Additional Editor Comments:

Dear Authors,

Both the reviewers have suggested major revision to the MS PONE-D-24-00993 entitled Dual RNA-Seq reveals transcriptome changes during Fusarium virguliforme -Trichoderma afroharzianum interactions. Please revise the comments carefully as soon as possible.

Reviewers' comments:

Reviewer's Responses to Questions

**Comments to the Author**

1. Is the manuscript technically sound, and do the data support the conclusions?

Reviewer #1: Yes

Reviewer #2: Partly

2. Has the statistical analysis been performed appropriately and rigorously? 

Reviewer #1: I Don't Know

Reviewer #2: Yes

3. Have the authors made all data underlying the findings in their manuscript fully available?

Reviewer #1: Yes

Reviewer #2: Yes

4. Is the manuscript presented in an intelligible fashion and written in standard English?

Reviewer #1: Yes

Reviewer #2: No

5. Review Comments to the Author

Reviewer #1: Dear Editor,

The MS entitled “Dual RNA-Seq reveals transcriptome changes during Fusarium virguliforme Trichoderma afroharzianum interactions” describes differences in transcriptome at different stages of their interaction. The authors have done a good piece of work and can be accepted after revision on the following points.

1. In Introduction Pg 3-4, paragraph 3, line 61 to 77 can be shortened.

2. Methodology is too long.

3. Why the expression of CAZymes and CBM domains proteins were upregulated?

4. What is the need of giving Figure 1?

5. Pg 9, Line 196, Where is figure 21?

6. The legends of figures can be improved.

7. Why the is written before scientific name of fungi?

8. How PCA was applied? Which tool or software was employed?

9. Venn diagrams are impressive but difficult to understand what the numbers signify.

10. There are too many figures. Only important figures must be included in the main MS and rest can be given as supplementary material.

11. How analysis of secreted proteins was done?

12. Discussion at some points appears vague. References including Differential Response of Extracellular Proteases of Trichoderma Harzianum Against Fungal Phytopathogens; and Elucidation of biocontrol mechanisms of Trichoderma harzianum against different plant fungal pathogens: Universal yet host specific response, may also be included in discussion part.

13. There are many grammatical, syntax and spelling errors independentely, the F. virguliforme, with aa similartrend, counts contrasted .

Reviewer #2: The MS PONE-D-24-00993 entitled “Dual RNA-Seq reveals transcriptome changes during Fusarium virguliforme - Trichoderma afroharzianum interactions” need major revision. The language of the MS also needs corrections. The details comments are described as;

Introduction is generalized. Authors should focus on role of Biological control agent and their mechanism. Trichoderma is a well-known biological control agent. Rephrase sentence line 38 to 42. Merger first and second paragraph. Font type and size in MS is not uniform. Remove The before the name of the genus. Follow uniform pattern for citation. Both number as well et al format has been used. Figure numbers are incorrect. It is not clear how authors separated mycelia for RNA and transcriptomics analysis during Fusarium-Trichoderma interaction.

Line 61 to 62: rephrase the sentence as it appears that Trichoderma may be responsible for seedling diseases.

Line 86 both T. afroharzianum isolates will occur better to write it as may occur.

Line 107 How An equal-sized plug were obtained? describe it.

Figure 1: why authors are giving pictorial representation? Instead use original figure only.

Line 131 to 133 rephrase the line.

Line 196 confronted with F. virguliforme, can be seen in figure 21 rephrase the sentence and figure numbers as figure 21 is wrong

It appears that figure 7 and figure 13 both represent interaction of Biocontrol agent –pathogen interaction. Then why it is repeated? Similarly of heat map and other analysis.

6. PLOS authors have the option to publish the peer review history of their article (what does this mean?). If published, this will include your full peer review and any attached files.

Reviewer #1: No

Reviewer #2: **Yes: **Rhydum Sharma

---

## [Author Response · Author response to Decision Letter 0]

30 Jun 2024

Response to Reviewers

Reviewer #1

Dear Editor,

The MS entitled "Dual RNA-Seq reveals transcriptome changes during Fusarium virguliforme Trichoderma afroharzianum interactions" describes differences in transcriptome at different stages of their interaction. The authors have done a good piece of work and can be accepted after revision on the following points.

1. In Introduction Pg 3-4, paragraph 3, line 61 to 77 can be shortened.

Response: The paragraph has been shortened as suggested.

2. Methodology is too long.

Response: Portions of the methodology have been abbreviated to reduce size. However, we feel that further reductions and omissions will lead to the deletion of crucial information needed to ensure the study's reproducibility.

3. Why the expression of CAZymes and CBM domains proteins were upregulated?

Response: As presented in the discussion, genes encoding CAZymes and CBM are known to be produced by antagonistic fungi and may play a major role in the breakdown of the fungal prey cell wall, enabling mycoparasitim.(starting line 564). 

4. What is the need of giving Figure 1?

Response: The authors believe that Figure 1 offers a valuable visual aid for readers to understand the different growth patterns of each Trichoderma isolate while interacting with F. virguliforme and where mycelia were harvested for RNA extraction. This figure was kept in the manuscript but moved to the supplemental material.

5. Pg 9, Line 196, Where is figure 21?

Response: We apologize for the typo. It should read Fig 3.

6. The legends of figures can be improved.

Response: Legends have been improved

7. Why the is written before scientific name of fungi?

Response: This issue has been corrected.

8. How PCA was applied? Which tool or software was employed?

Response: PCA was applied using DESeq2 (line 172). We included that DESeq2 was used within R for the data visualization.

9. Venn diagrams are impressive but difficult to understand what the numbers signify.

Response: We appreciate the feedback. The objective of the Venn diagram was to show at a glance the counts of DEG that were shared (or unique) in the transcriptome of each fungus. Unfortunately, with our large dataset (we have as a dual-RNAseq featuring 3 different interaction times), this was the best representation we envisioned without including too many figures. The main points of the Venn diagrams are explained in the text.

10. There are too many figures. Only important figures must be included in the main MS and rest can be given as supplementary material.

Response: Several figures and tables were moved to Supplementary material, as follows:Table 1 = Table S1

Table 2 = Table S2

Figure 1 = Figure S1

Figure 5 = Figure S3

Table 3= Table S3

Figure 11 = Figure S4

11. How analysis of secreted proteins was done?

Response: The secreted proteins profile was done based on the DEG that were categorized as "GO Category Extracellular" in the enrichment analysis. This information was added to the material and methods section.

12. Discussion at some points appears vague. References including Differential Response of Extracellular Proteases of Trichoderma Harzianum Against Fungal Phytopathogens; and Elucidation of biocontrol mechanisms of Trichoderma harzianum against different plant fungal pathogens: Universal yet host specific response, may also be included in discussion part.

Response: The discussion has been improved based on the suggestions of the reviewer. 

13. There are many grammatical, syntax and spelling errors independentely, the F. virguliforme, with aa similartrend, counts contrasted .

Response: We appreciate the reviewers' feedback on this matter. The manuscript has been carefully revised to eliminate grammatical, syntax, and spelling errors. 

Reviewer #2

The MS PONE-D-24-00993 entitled "Dual RNA-Seq reveals transcriptome changes during Fusarium virguliforme - Trichoderma afroharzianum interactions" need major revision. The language of the MS also needs corrections. The details comments are described as;

The introduction is generalized. Authors should focus on role of Biological control agent and their mechanism. Trichoderma is a well-known biological control agent. 

Response: The introduction has been improved based on the reviewer's suggestion. However, a large focus is still on Trichoderma, as this research was focused on determining the underlying differences between unique phenotypes of 2 Trichoderma isolates while interacting with Fusarium virguliforme.

Rephrase sentence lines 38 to 42. 

Response: The sentence has been rephrased.

Merger first and second paragraph. 

Response: Paragraphs were merged

Font type and size in MS is not uniform. Remove The before the name of the genus. 

Response: These issues have been corrected.

Follow uniform pattern for citation. Both number as well et al format has been used. Figure numbers are incorrect. It is not clear how authors separated mycelia for RNA and transcriptomics analysis during Fusarium-Trichoderma interaction.

Response: The information about how mycelia were collected from the interactions has been added to the manuscript. Citations have been fixed to the same pattern, and all figure numbers have been corrected. We thank the reviewers for bringing these issues to our attention.

Line 61 to 62: rephrase the sentence as it appears that Trichoderma may be responsible for seedling diseases.

Response: The sentence has been rephrased

Line 86 both T. afroharzianum isolates will occur better to write it as may occur.

Response: This suggestion has been incorporated into the sentence. 

Line 107 How An equal-sized plug were obtained? describe it.

Response: Equal-sized plugs were obtained by using a sterile glass pipette to cut the mycelial plugs from actively growing colonies. Therefore, all plugs were 6 mm in diameter. This information has been included as requested.

Figure 1: why authors are giving pictorial representation? Instead use original figure only.

Response: Taking high-quality pictures of Petri dishes is difficult due to the natural glare of light reflecting on the plate lid. The glare can be reduced by removing the lid, but that would have compromised the plates at the earlier stages of the interaction. Therefore, the schematic was created by the authors as a visual aid for readers to understand the different growth patterns of each Trichoderma isolate while interacting with F. virguliforme, and where mycelia were harvested for RNA extraction. The schematic was transferred to the supplementary material.

Line 131 to 133 rephrase the line.

Response: The sentence has been restructured.

Line 196 confronted with F. virguliforme, can be seen in figure 21 rephrase the sentence and figure numbers as figure 21 is wrong

Response: The sentence was rephrased, and the Figure number was corrected (Figure 3). We apologize for the typo.

It appears that figure 7 and figure 13 both represent interaction of Biocontrol agent –pathogen interaction. Then why it is repeated? Similarly of heat map and other analysis.

Response: In this dual-RNAseq study, we looked at the transcriptomes of F. virguliforme (Fv) and each of the biocontrol agents separately. Figure 7 shows DEG genes in the F. virguliforme transcriptome, while Figure 13 shows DEG genes from the T. afroharzianum transcriptome. They are not repetitions. The manuscript is structured in a way that shows the F. virguliforme transcriptome first (Fig 4 to 9), and then we proceed to show the T. afroharzianum transcriptome (Fig 10 to 15).

---

## [Editor Report · Decision Letter 1]

8 Sep 2024

Dual RNA-Seq reveals transcriptome changes during Fusarium virguliforme -Trichoderma afroharzianum  interactions

PONE-D-24-00993R1

Dear Dr. Rocha,

We’re pleased to inform you that your manuscript has been judged scientifically suitable for publication and will be formally accepted for publication once it meets all outstanding technical requirements.

Kind regards,

Erika Kothe

Academic Editor

PLOS ONE

---

## [Editor Report · Acceptance letter]

23 Sep 2024

PONE-D-24-00993R1 

PLOS ONE

Dear Dr. Rocha, 

I'm pleased to inform you that your manuscript has been deemed suitable for publication in PLOS ONE. Congratulations! Your manuscript is now being handed over to our production team.

Kind regards, 

on behalf of

Prof. Dr. Erika Kothe 

Academic Editor

PLOS ONE